# Q-Factor Enhancement of Coupling Bragg and Local Resonance Band Gaps in Single-Phase Phononic Crystals for TPOS MEMS Resonator

**DOI:** 10.3390/mi13081217

**Published:** 2022-07-29

**Authors:** Lixia Li, Weitao He, Zhixue Tong, Haixia Liu, Miaoxia Xie

**Affiliations:** 1School of Mechanical and Electrical Engineering, Xi’an University of Architecture and Technology, Xi’an 710055, China; lilixia@xauat.edu.cn (L.L.); hwt0087@xauat.edu.cn (W.H.); ilysay@xauat.edu.cn (H.L.); xiemiaoxia@xauat.edu.cn (M.X.); 2Institute of Mechanics, Xi’an University of Architecture and Technology, Xi’an 710055, China

**Keywords:** anchor loss, MEMS resonator, phononic crystals, single-phase material, coupling band gap, quality factor

## Abstract

This paper presents a type of single-phase double “I” hole phononic crystal (DIH-PnC) structure, which is formed by vertically intersecting double “I” holes. By using the finite element method, the complex energy band curve, special point mode shapes, and different delay lines were calculated. Numerical results showed that DIH-PnC yielded ultra-wide band gaps with strong attenuation. The formation mechanism is attributed to the Bragg-coupled local resonance mechanism. The effects of the pore width in DIH-PnC on the band gaps were further explored numerically. Significantly, as the pore width variable, the position of the local resonance natural frequency could be modulated, and this enabled the coupling between the local resonance and the Bragg mechanism. Subsequently, we introduced this DIH-PnC into the thin-film piezoelectric-on-silicon (TPOS) resonator. The results illustrated that the anchor loss quality factor ($Q_{\mathrm{anc}}$) of the DIH-PnC resonator was 20,425.1% higher than that of the conventional resonator and 3762.3% higher than the $Q_{\mathrm{anc}}$ of the cross-like holey PnC resonator. In addition, the effect of periodic array numbers on $Q_{\mathrm{anc}}$ was researched. When the $Q_{\mathrm{anc}}$ reached 1.12 × 10^6^, the number of the period array in DIH-PnC only needed to be 1/6 compared with cross-like holey PnC. Adopting the PnC based on the coupling Bragg and local resonance band gaps had a good effect on improving the $Q_{\mathrm{anc}}$ of the resonator.

## 1. Introduction

With the development of microelectromechanical systems (MEMS) technology, thin-film piezoelectric-on-silicon (TPOS) is considered by most researchers to be a miniaturized, preferred choice for high-performance, low-power integrated resonators [1]. Regardless of the application in sensors, detectors, and small acoustic antennas, TPOS resonators are required to have a high quality factor (Q) so that higher accuracy can be achieved [2]. At present, scholars have carried out a lot of research on the Q of TPOS resonators [3,4,5,6]. The research shows that one of the main factors affecting the low Q of TPOS resonators is that when the resonator is working, elastic waves propagate acoustic energy to the resonator substrate through the anchor point support, resulting in an unavoidable anchor loss [7].

On this basis, some scholars have proposed various methods to reduce anchor point loss [8,9,10,11,12,13,14,15,16,17]: Harrington et al. demonstrated an arc-shaped acoustic reflector in aluminum nitride on a silicon LVR, enabling the reflection of acoustic energy back to the resonator [8]. Di et al. improved the Q value by using grooves and convex curve pairs at the edge of the resonator to reduce anchoring losses, increasing the Q of the TPOS resonator by more than nine times [9]. Another common method to form acoustic reflectors is to rely on the band gap properties of phononic crystals (PnCs), which can effectively block the propagation of acoustic energy within the band gap range [10]. Zhu et al. investigated a two-dimensional air-hole PnC unit cell in a 143 MHz TPOS resonator which could double the Q [11]. Apart from air-holes [11], other shapes of the PnC unit cell, such as rings [12], cross-like holes [13], fractals [14], snowflakes [15], solid discs [16], and spiderweb-like [17], have been also reported. Among them, the research of Ardito et al. showed that a larger band gap width is beneficial to confine elastic waves, thereby improving the Q [16]. The above-mentioned PnC structures are all based on the band gap generated by the Bragg scattering mechanism (BG). Since the Bragg scattering mechanism is mainly formed by the multiple scattering of the periodic structure and the interference effect between the structural materials, the attenuation degree has a strong requirement on the periodicity of the structure [18,19]. Therefore, TPOS resonators made of PnC based on the Bragg mechanism tend to have a large area occupied by PnC [11,12,13,14,15,16,17].

It is well-known that the local resonance mechanism (LR) of acoustic metamaterial was proposed by Liu et al. in 2000. Compared with the Bragg mechanism, this local resonance mechanism has the advantages of a lower band gap, stronger attenuation, and lower periodicity requirements [20,21]. In recent years, the coupling mechanism between Bragg and local resonance has attracted the attention of a large number of researchers [18,19,22,23,24,25,26,27]. Bo Yuan et al. achieved Bragg-coupled local resonance band gaps (LR BGs) by tuning the local resonant band gap to coincide with the Bragg band gap in multi-materials [18]. A. O. Krushynska et al. achieved quasi-LR BGs in single-phase materials [19]. LR BGs can not only widen the band gap but also enhance the attenuation performance of the PnC [18]. Xiangyu Tian et al. realized LR BGs at low frequencies with perforated PnC [26]. Qiang Wang et al. obtained LR BGs on a periodic “sandwich” plate type [27]. Currently, only Yinjie Tong et al. in the field of TPOS resonators proposed multi-material pillar-based phononic crystals that may be based on coupling mechanisms [28]. However, the preparation and application of multi-material phononic crystals are limited. Therefore, there is a need for a single-material coupling mechanism phononic crystal that can be applied to TPOS resonators.

In this paper, a single-phase DIH-PnC structure is proposed, which was formed by vertically intersecting double “I” holes. In the second part, combined with the complex energy band curve and the special point mode shape, the band gap mechanism of the DIH-PnC structure is deeply analyzed and the attenuation degree is analyzed by three delay line transmission parameters $S_{21}$ and the normalized displacement field, and finally, the hole parameter pair is analyzed for the effect of the band gap mechanism. In the third section, the Q effect on the resonator using the coupled band gap mechanism and the number of periodic arrays of PnC is analyzed by introducing this single-phase DIH-PnC structure into the TPOS resonator. Finally, a brief conclusion is arranged.

## 2. Phononic Crystal Design

### 2.1. Dispersion Relations

In this study, a double “I” hole (DIH-PnC: double “I” hole phononic crystal) is proposed. The structure consists of vertically intersecting double “I” holes forming the same four-square masses (as shown in Figure 1a). Figure 1a is the structure of DIH-PnC, and Figure 1b is the unit cell of the original cross-like holey PnC. The right side of Figure 1 is the corresponding unit cell model. Both the DIH-PnC structure and the cross-like holey PnC structures are all square lattices and show a symmetrical form. Therefore, DIH-PnC is identical to the irreducible Brillouin region of the cruciform PnC structure (as shown in Figure 1). Among them, the geometric parameters include the lattice constant of the unit cell a=24 µm, the height h=10 µm, the length of the inner hole m=11 µm and n=20 µm, and the width of the inner hole c and d. Different internal mass block sizes and cross-like holey PnC structures can be achieved by adjusting the parameters of different internal hole widths c and d. The material used in the DIH-PnC is consistent with the TPOS resonator substrate material and is made of anisotropic single-crystal silicon that receives the resonator to exhibit higher power handling capabilities. The default *x*, *y*, and *z* axes of anisotropic single-crystal silicon is set to (110), (−110), and (001) directions of the standard direction (100) for silicon wafers, and the elastic modulus E of anisotropic single-crystal silicon, shear modulus G, and Poisson’s ratio σ are shown in Table 1.

In order to analyze the band gap properties of the PnC structure, the Bloch–Floquet theoretical calculation method based on the finite element method was used [29,30]. From the real wave vector energy band curve and the imaginary wave vector attenuation curve combined with the band gap boundary displacement field, three aspects were studied. Due to the Bloch–Floquet theoretical calculation, this method was based on the infinite periodic structural arrangement of unit cells along the x and y directions, so we only needed to study the unit cell model shown on the right side of Figure 1. The periodic boundary conditions applied to the x and y directions of the unit cell structure are expressed as:(1)$$p\left(R+a\right)=p\left(R\right)e^{iKa},$$
where *R* is the direction vector; *a* is the unit cell lattice constant; *K* is the wave vector.

The eigenvalue problem was solved by sweeping the wave vector *K* (real wave vector, imaginary wave vector) over the boundary of the irreducible Brillouin region of Figure 1.

When the inner hole width is d=3.5 µm and c=1 µm, the complex energy band diagram of the DIH-PnC structure is shown in Figure 2a. For the convenience of viewing, the complex energy band within 20–26 MHz is partially enlarged. The solid blue line and the red dotted line represent the real wave vector energy band curve and the imaginary wave vector attenuation curve of the wave vector *K*, respectively. The right side of Figure 2a shows the mode shape diagram of the special point of the first band gap boundary, and the left side of Figure 2a shows the mode shape diagram of the special point of the second band gap boundary. It can be seen from the complex energy band curve in Figure 2a that there were two complete band gaps in the frequency range of 0–160 MHz, of which the first band gap was at 22.9−24.9 MHz and the second was at 77.6–125.1 MHz. It is worth noting that there was a flat band in the two band gap ranges and a third flat band appeared at 131.5 MHz. The special point mode shape of the flat band is shown in Figure 2b. When the structure is known to be at the natural frequency of the local resonance, the virtual wave vector decay curve exhibits obvious sharp features [31]. Combining with the attenuation curve of the virtual wave vector where the flat band is located in Figure 2, it could be known that the flat band is caused by the first natural frequency, the second natural frequency, and the third natural frequency of the four mass blocks. It can be seen from the special point mode shapes of A/B in Figure 2 that the first band gap boundary mode shapes all show the torsional local resonance mode of the diagonal mass block, and opening the first band gap was the local resonance band gap: 22.9−24.9 MHz.

In order to further analyze the second band gap mechanism of the DIH-PnC structure, the band gap mechanism of the cross-like holey PnC structure formed by comparing internal pore width parameters c=m=11 µm and d=n=20 µm was examined. The complex energy band curve is shown in Figure 3, and the right side of Figure 3 is the mode shape diagram of the special point of the boundary of the band gap. It can be seen from the complex energy band curve in Figure 3 that there was a complete band gap of 101–128 MHz in the cross-like holey PnC structure in the range of 0–160 MHz, and the virtual wave vector attenuation curve showed a continuous and stable change in the range of the band gap. The mode shape diagram on the right side of Figure 3 shows that the outer frame of the band gap boundary showed a symmetrical mode. Therefore, the band gap 101–128 MHz of the cross-like holey PnC structure is the Bragg band gap mechanism. In Figure 2a, the mode shape of the C/D point is different from that of A/B in Figure 3; the mode shape of C in Figure 2 point shows the interaction between the torsional mode shape of the inner mass and the symmetrical mode shape of the outer frame. D in Figure 2 point mode shapes show the interaction of the outer frame antisymmetric mode shapes with a small number of inner mass torsional modes. Meanwhile, in Figure 2a, the attenuation curve of the virtual wave vector in the second band gap is stronger than that of the virtual wave vector in the band gap in Figure 3. Therefore, the second band gap is formed by Bragg and the local resonance. Compared with the 27 MHz band gap width of the cross-like holey PnC, the second band gap of the DIH-PnC structure reached 47.5 MHz, and the band gap was widened by 175%. At the same time, the DIH-PnC structure had a stronger attenuation ability in the second band gap than the attenuation ability in the cross-like holey PnC band gap. The results showed that the coupling mechanism based on Bragg and local resonance could widen the band gap and improve the attenuation capability.

### 2.2. Transmission Spectrum

In order to verify the stopband effect of the coupling mechanism DIH-PnC structure under finite arrays, we used different delay line models for comparative analysis and used the transmission parameter (i.e., $S_{21}$) to measure the degree of stop band. Specifically, the delay line model was established as shown in Figure 4; Figure 4a is the solid structure comparison delay line model; Figure 4b is the cross-like holey PnC structure comparison delay line model; Figure 4c is the DIH-PnC structure delay line model; among them, the PnC adopted a 2-row × 5-column array. As shown in Figure 4, an x-direction displacement excitation is applied at the input probe, and the displacement result is picked up using the output probe. Furthermore, in order to reduce the interference of elastic wave reflection on the $S_{21}$, perfect matching layers (PMLs) are set at both ends of the model, imposing periodic boundary conditions on both sides of the delay line.

The $S_{21}$ is measured in decibels from the input probe to the output probe and is expressed as:(2)$$S_{21}\left(\mathrm{dB}\right)=10log_{10}\left(\frac{S_{0}}{S_{1}}\right),$$

Among them, $S_{0}$ represents the output displacement and $S_{1}$ represents the input displacement.

The result of delay line $S_{21}$, as shown in Figure 5, in the band gap frequency range, was that the delay line composed of PnC had a larger drop than the solid contrast delay line. When the frequency was in the range of 24–27 MHz, the $S_{21}$ of the DIH-PnC delay line had a relatively obvious attenuation and exhibited a sharp characteristic, which is consistent with the analysis in Section 2.1, due to the local resonance band gap. When the frequency was 75–125 MHz, although the $S_{21}$ of the cross-like holey PnC delay line and the $S_{21}$ of the DIH-PnC delay line had obvious attenuation, the overall attenuation of the DIH-PnC delay line was more obvious than that of the cross-like holey PnC delay line. When the frequency was at 128 MHz, the transmission coefficient $S_{21}$ of the cross-like holey PnC delay line reached the maximum attenuation of −62 dB. When the frequency was at 120 MHz, the transmission coefficient $S_{21}$ of the DIH-PnC delay line reached the maximum attenuation of −100 dB. It can be seen that the maximum attenuation of the DIH-PnC delay line was 1.6 times that of the cross-like holey PnC delay line. The attenuation effect of the DIH-PnC delay line was weakened at 110 MHz, which was caused by the second flat band in the second band gap of the DIH-PnC structure.

To further analyze the stop band situation within 75–125 MHz, the normalized displacement field of different delay lines at the same frequency of 108 MHz is plotted using a heat map (as shown in Figure 6); red indicates maximum displacement, white indicates no movement, and arrows indicate the direction of displacement at each point. Figure 6a shows the normalized displacement field of the solid contrast delay line, Figure 6b is the normalized displacement field of the cross-like holey PnC delay line, and Figure 6c is the normalized displacement field of the delay line of the DIH-PnC structure. It can be seen from the figure that when the frequency was at 108 MHz, the normalized displacement field of the cross-like holey PnC delay line gradually weakened as the elastic wave propagated in the x-direction. When the frequency was at 108 MHz, the DIH-PnC structure delay line not only had the normalized displacement field but gradually weakened as the elastic wave propagated in the x-direction. At the same time, a large number of elastic waves were blocked on the left side of the first column of DIH-PnC unit cells due to the local resonance of the internal mass. It was further verified that the second band gap of the DIH-PnC structure is based on the coupling mechanism of Bragg and local resonance.

### 2.3. Influence of Hole c Parameter

To analyze the effect of pore width on the proposed single-phase band gap mechanism, by keeping other parameters unchanged, the complex energy band curve was studied when only the hole width c was changed (as shown in Figure 7). Figure 7a–c shows the real wave vector energy band curves when c=1 µm, c=3 µm, and c=5 µm.

It can be seen from Figure 7 that when c=1 µm, the second band gap of the DIH-PnC structure was 77.6–125.1 MHz. When c=3 µm, the second band gap of the DIH-PnC structure: 85.3–125.0 MHz. When c=5 µm, the second band gap of the DIH-PnC structure: 94.1–128 MHz.

As shown in Figure 7a, when c=1 µm, there were three flat bands caused by the third-order natural frequencies ($f_{1}$, $f_{2}$*,* and $f_{3}$) in the range of 0–150 MHz. The second flat band was located in the second band gap, the third flat band was located at the boundary of the second band gap, and the second band gap mechanism was a strong coupling mechanism. The virtual wave vector decay curve in the second band gap had the largest absolute value at 102.3 MHz (|Im(q)|). As shown in Figure 7b, when c=3 µm, only two flat bands appeared in 0–150 MHz, and the third flat band disappeared in 0–150 MHz. The second flat band (generated by the second natural frequency $f_{2}$) moved to the boundary of the second band gap, and the second band gap mechanism was the coupled band gap mechanism. The absolute value of the virtual wave vector attenuation curve in the second band gap was larger at 107.4 MHz (|Im(q)|). It is particularly noteworthy that when the second flat band moved to the second boundary, it interacted with the original band gap boundary energy band, so the second flat energy band exhibited a bending phenomenon. As shown in Figure 7c, when c=5 µm, only one flat band remained in 0–150 MHz. At this time, the flat band produced by the third-order natural frequency was far from the boundary of the second-order band gap. The second band gap mechanism reverted to the Bragg gap. The absolute value of the virtual wave vector decay curve in the second band gap was smaller at 114.7 MHz (|Im(q)|).

Therefore, by adjusting the pore width parameter 𝑐, the position of the local resonance band gap generated by the natural frequency could be adjusted, and the coupling between the local resonance mechanism and the Bragg mechanism could be adjusted.

## 3. Resonators Design

### 3.1. Resonator Model

The resonators designed in this paper are all TPOS resonators. A simplified model of a conventional resonator for TPOS is shown in Figure 8, where the silicon dioxide layer of the silicon-on-insulator substrate and the anchor substrate are omitted.

The width extension (WE) vibration mode expression of the TPOS resonator is [32]:(3)$$f_{r}=\frac{nv}{2W_{r}}$$

In Equation (3), v represents the speed of sound in the resonator, $W_{r}$ is the width of the rectangular resonator, and n represents that the resonator has a harmonic mode of n order. In this paper, the rectangular resonator n was of 5th order. Design resonant frequency $f_{r}=107.6\;\mathrm{MHz}$, resonator width $W_{r}=197\;µm$, resonator length $L_{r}=592\;µm$, support beam length $L_{x}=59.2\;µm$, and support beam width $W_{x}=20\;µm$. The thickness of the base silicon was 10 µm, and the thickness of the electrodes and the piezoelectric layer was 0.5 µm. The power is input through the electrodes Al at the upper and lower ends of the resonator in Figure 8, and the piezoelectric layer AlN performs forward and reverse piezoelectric effects to drive the vibration of the base plate. Finally, the middle electrode of the resonator is used for output power. A perfectly matched layer ($L_{PML}=3\times \lambda$) is used on the outside of the resonator to completely absorb the reflection of the acoustic wave and avoid the reflected wave from affecting the TPOS resonator.

In order to verify the effect of the DIH-PnC resonator on anchor loss, we established a comparative analysis of different types of resonators. A conventional resonator is shown in Figure 9a, Figure 9b is a resonator using the cross-like holey PnC shown in Figure 3, and Figure 9c is a resonator using the DIH-PnC shown in Figure 2. Among them, a PnC array with six rows and two columns was used on the substrate.

The effect of the coupling mechanism DIH-PnC applied on the TPOS resonator to prevent sonic flooding can be verified by the anchor loss quality factor ($Q_{\mathrm{anc}})$ [33,34]. The $Q_{\mathrm{anc}}$ of the TPOS resonator in the WE vibration mode can be calculated by the finite element calculation method. $Q_{\mathrm{anc}}$ can be expressed as [35]:(4)$$Q_{\mathrm{anc}}=\frac{Re\left(\omega \right)}{2lm\left(\omega \right)}$$
where Re(ω) represents the real part of the resonant angular frequency of the resonator, and lm(ω) represents the imaginary part of the resonant angular frequency of the resonator.

Figure 10 shows the mode shapes and $Q_{\mathrm{anc}}$ of the three types of resonators at the resonant frequency. Figure 10a $Q_{\mathrm{anc}}$ = 60,565 for conventional resonator, Figure 10b $Q_{\mathrm{anc}}$ = 321,850 for cross-like holey PnC resonator, and Figure 10c $Q_{\mathrm{anc}}$ = 12,431,000 for DIH-PnC resonator. It can be seen from this that the $Q_{\mathrm{anc}}$ of the DIH-PnC resonator was 20,425.1% higher than that of the conventional resonator and 3762.3% higher than the $Q_{\mathrm{anc}}$ of the cross-like holey PnC resonator. Therefore, under the same PnC array, the DIH-PnC structure with coupled band gap mechanism could achieve higher $Q_{\mathrm{anc}}$.

### 3.2. Optimization of Phononic Crystals Array

In order to further analyze the influence of PnC array layout on $Q_{\mathrm{anc}}$, different array parameters were investigated. A full parametric scan was performed using an m-row n-column PnC array layout. Specifically, it took values two, four, and six for m rows and two, three, and four for n columns. The results of $Q_{\mathrm{anc}}$ are shown in Figure 11, and the specific data are shown in Table 2.

It can be seen from Figure 11 that the $Q_{\mathrm{anc}}$ of the DIH-PnC resonator was much higher than that of the conventional resonator and the cross-like holey PnC resonator. For DIH-PnC resonators, $Q_{\mathrm{anc}}$ did not increase with the number of periodic arrays. When the PnC array was in six rows × two columns, the $Q_{\mathrm{anc}}$ of the DIH-PnC resonator reached a maximum value of 1.24 × 10^7^. For the cross-like holey PnC resonator, $Q_{\mathrm{anc}}$ gradually increased with the increase in the number of periodic arrays. When the PnC array was six rows × four columns, the $Q_{\mathrm{anc}}$ of the cross-hole PnC resonator reached the maximum value of 1.12 × 10^6^. However, if the $Q_{\mathrm{anc}}$ of the resonator reached 1.12 × 10^6^, the DIH-PnC resonator only needed to use a two rows × two columns PnC periodic array, and the periodic array number was only 1/6 compared with the cross-like holey PnC.

Therefore, when the DIH-PnC resonator based on the coupling mechanism reached a higher value of $Q_{\mathrm{anc}}$, the number of periodic arrays was less, thereby reducing the area occupied by the PnC in the substrate.

## 4. Conclusions

In this paper, a DIH-PnC structure composed of vertically intersecting duplex holes was investigated, and the inter-coupling band gap of the Bragg mechanism and the local resonance mechanism was obtained in a single-phase material. Through the analysis of the complex energy band and different delay lines, it was known that the DIH-PnC structure could achieve an ultra-wide band gap: 77.6–125.1 MHz, which was 175% wider than the cross-like holey PnC band gap. Through the wave field analysis of the special point mode shape and delay line at 108 MHz, it was concluded that the ultra-wide band gap of the DIH-PnC structure was generated by the mutual coupling of Bragg and local resonance. Moreover, the maximum attenuation of the DIH-PnC delay line was 1.6 times that of the cross-like holey PnC delay line. The evolution of the band gap mechanism with the change of the hole width c was studied, and the second band gap mechanism transformed from the Bragg–local resonance coupled mechanism to a separate Bragg mechanism as c increased. The $Q_{\mathrm{anc}}$ of the TPOS resonator in the WE vibration mode could be calculated by the finite element calculation method. When the number of PnC periodic arrays was six rows × two columns, the $Q_{\mathrm{anc}}$ of the DIH-PnC resonator could reach 1.24 × 10^7^, which was 20,425.1% higher than that of the conventional resonator and 3762.3% higher than that of the cross-like holey PnC resonator. In addition, the effect of different PnC periodic array numbers on $Q_{\mathrm{anc}}$ was studied. When the $Q_{\mathrm{anc}}$ of the cross-like holey PnC resonator reached the maximum value of 1.12 × 10^6^, six rows × four columns of the periodic array number were needed, while for the DIH-PNC resonator, as $Q_{\mathrm{anc}}$ was 1.12 × 10^6^, the periodic array number was only 1/6 compared with the cross-like holey PnC.

Therefore, DIH-PnC using a coupling-based mechanism had an ultra-wide band gap, strong attenuation capability, and low periodic dependence and was fabricated from a single material. Moreover, when applied to the TPOS resonator, the $Q_{\mathrm{anc}}$ value could be guaranteed to be high, and the number of periodic arrays reaching PnC was small, thereby reducing the area occupied by PnC in the substrate of the TPOS resonators.

## Figures and Tables

**Figure 1 micromachines-13-01217-f001:**
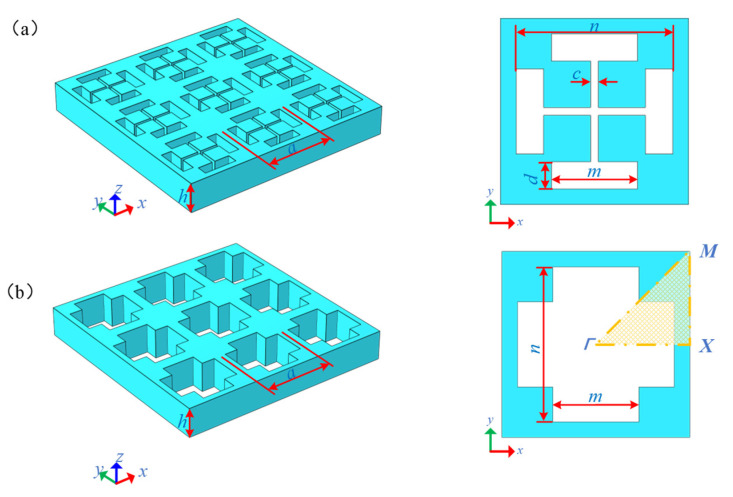
DIH-PnC structure: (**a**) DIH-PnC structure, (**b**) cross-like holey PnC structure. The right side is a single cell.

**Figure 2 micromachines-13-01217-f002:**
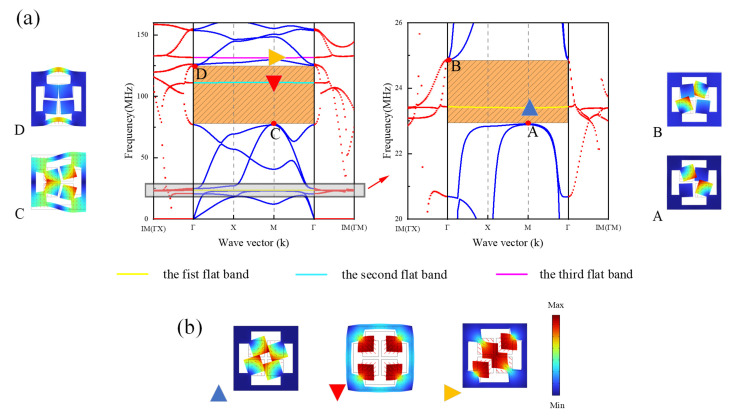
DIH-PnC structure, (**a**) is the complex energy band curve in the range of 0–160 MHz, the right side is the special point mode shape of the first band gap boundary, and the left side is the second band gap special point mode shape. The blue solid line represents the real wave vector energy band curve, and the red dotted line represents the imaginary wave vector attenuation curve. (**b**) Is the 0–160 MHz inside flat band with special point mode shape.

**Figure 3 micromachines-13-01217-f003:**
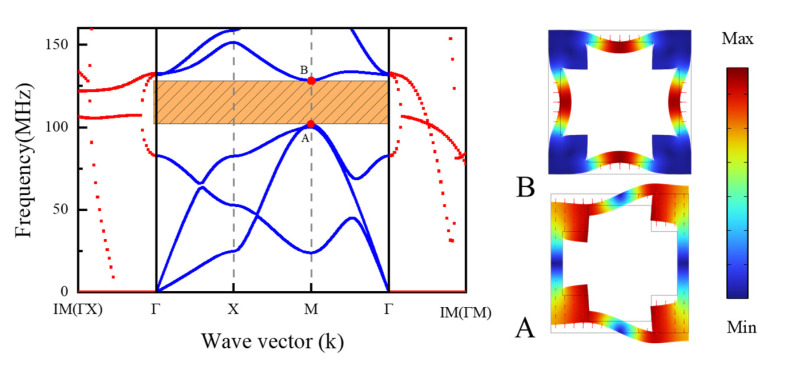
Cross-like holey PnC structure. The left side is the complex energy band curve of the cross-like holey PnC structure, the blue solid line represents the real wave vector energy band curve, and the red dotted line represents the imaginary wave vector attenuation curve. On the right is the special point mode shape of the band gap boundary.

**Figure 4 micromachines-13-01217-f004:**
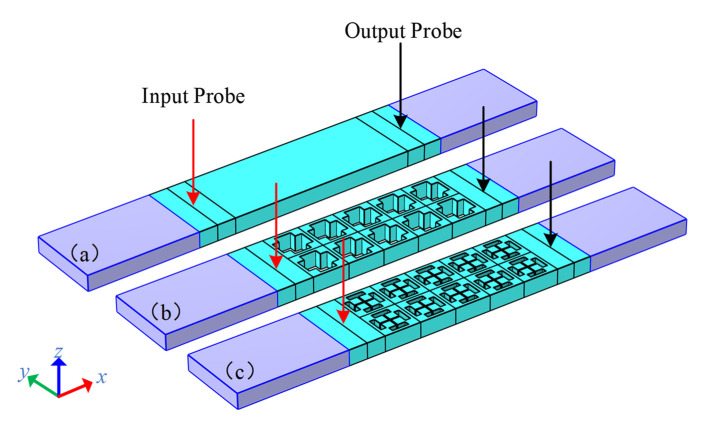
Delay line models: (**a**) solid versus delay line, (**b**) cross-like holey PnC structure versus delay line, and (**c**) DIH-PnC delay line. The deep areas at both ends are PML areas. The red arrows represent the input displacement excitation, and the black arrows represent the picked displacement results.

**Figure 5 micromachines-13-01217-f005:**
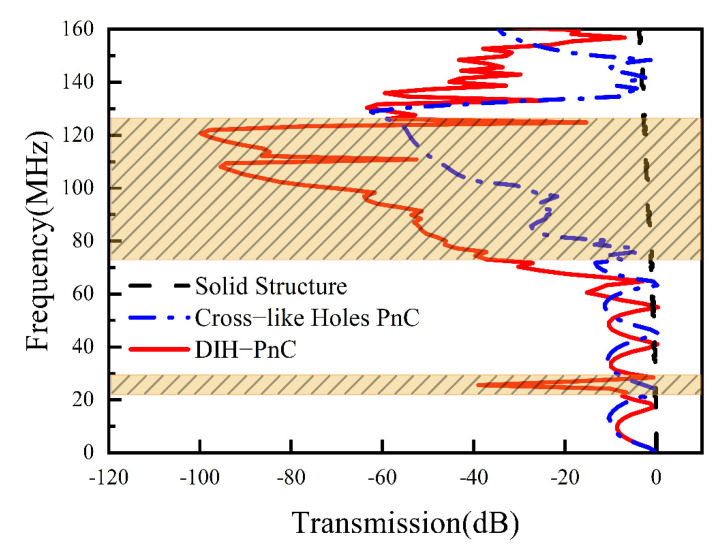
Delay line $S_{21}$.The gray dashed line is the solid contrast delay line, the blue dotted line is the cross-like holey PnC delay line, and the red solid line is the DIH-PnC delay line. The shaded area is the band gap range in the x-direction.

**Figure 6 micromachines-13-01217-f006:**
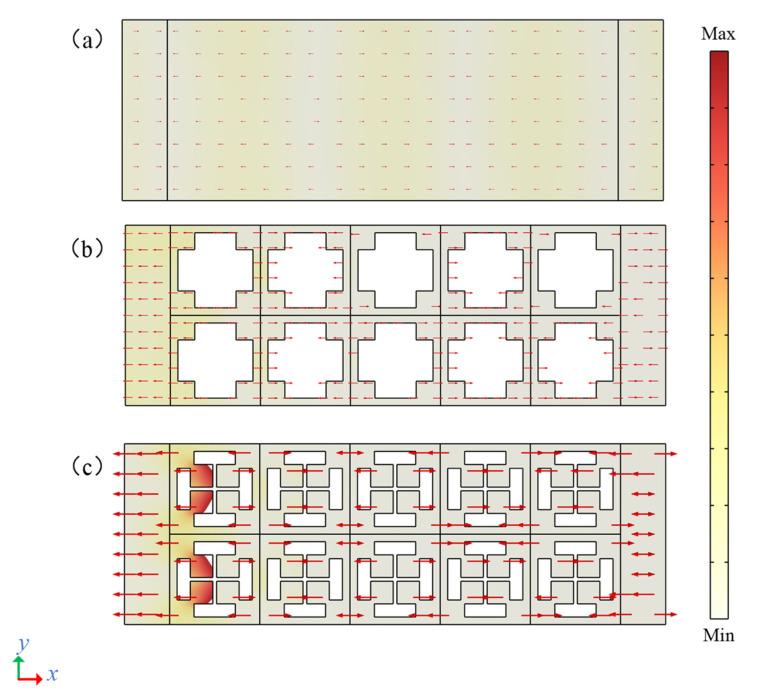
Normalized displacement field of the delay line model at 108 MHz. (**a**) Solid contrast delay line, (**b**) cross-shaped delay line, (**c**) DIH-PnC delay line.

**Figure 7 micromachines-13-01217-f007:**
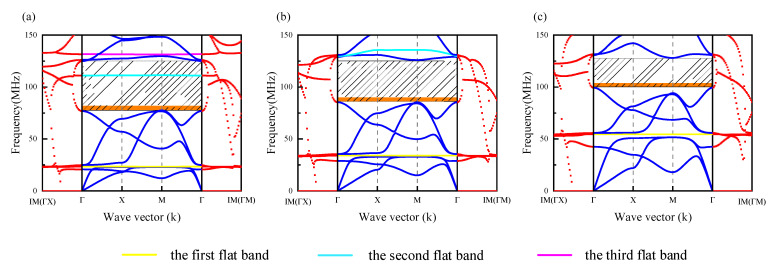
Complex energy band curve with different hole widths, The blue solid line represents the real wave vector energy band curve, and the red dotted line represents the imaginary wave vector attenuation curve. (**a**) c=1 µm, (**b**) c=3 µm, and (**c**) c=5 µm.

**Figure 8 micromachines-13-01217-f008:**
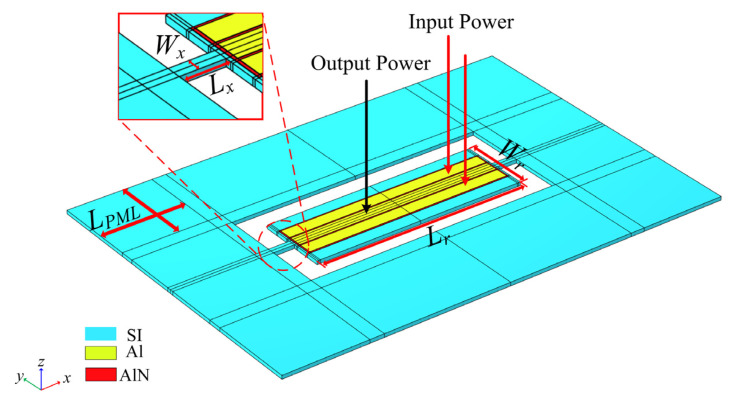
Simplified model of a conventional resonator for TPOS.

**Figure 9 micromachines-13-01217-f009:**
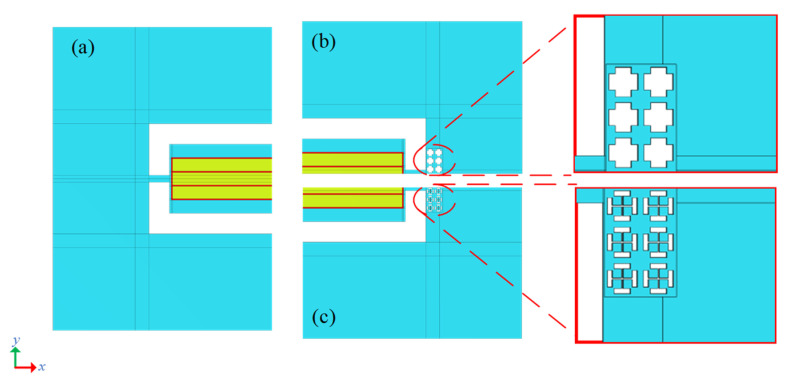
Simplified model for resonator simulation. (**a**) Model of a conventional TPOS resonator, shown as 1/2 the size of a conventional resonator. (**b**) 1/4 of the cross-like holey PnC resonator, (**c**) 1/4 of the DIH-PnC resonator. Among them, the PnC adopted an array of 6 rows × 2 columns.

**Figure 10 micromachines-13-01217-f010:**
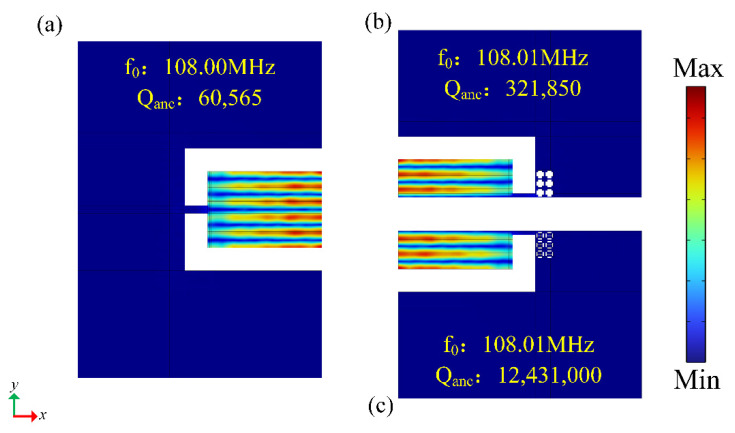
Resonator simulation results. (**a**) Conventional TPOS resonator model, shown as 1/2 of the conventional resonator. (**b**) 1/4 of the cross-like holey PnC resonator, (**c**) with DIH-PnC resonance 1/4 of the device. $f_{0}$ represents the resonant frequency and $Q_{\mathrm{anc}}$ represents the anchor loss quality factor.

**Figure 11 micromachines-13-01217-f011:**
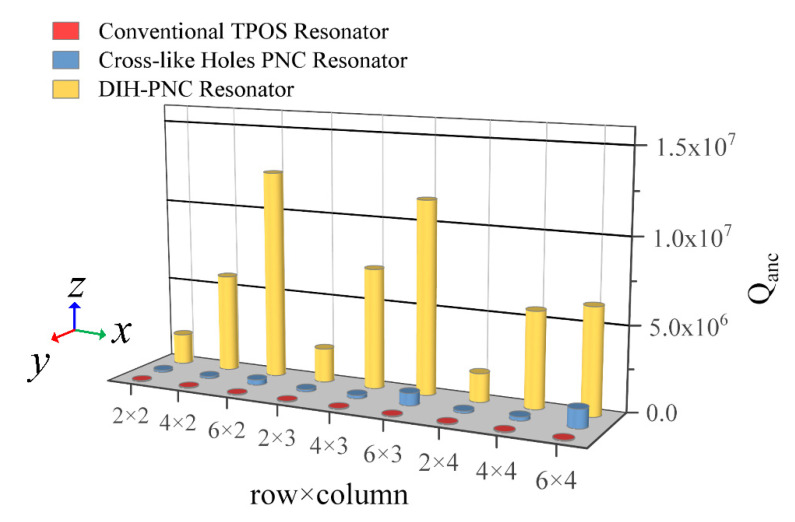
$Q_{\mathrm{anc}}$ values under different rows × columns. The *x*-axis represents different PnC rows × columns layouts, the *y*-axis represents the different resonator, and the *z*-axis represents the $Q_{\mathrm{anc}}$ values. Red cylinders represent conventional resonators, blue cylinders represent cross-like holey PnC resonators, and yellow cylinders represent DIH-PnC resonators.

**Table 1 micromachines-13-01217-t001:** Properties of Materials.

Elastic Modulus	Shear Modulus	Poisson’s Ratio
E_x_ = 169 GPa	G_xy_ = 50.9 GPa	σ_xy_ = 0.064
E_y_ = 169 GPa	G_yz_ = 79.6 GPa	σ_yz_ = 0.36
E_z_ = 130 GPa	G_zx_ = 79.6 GPa	σ_zx_ = 0.28

**Table 2 micromachines-13-01217-t002:** $Q_{\mathrm{anc}}$ values under different arrays.

Row × Column	Conventional TPOS Resonator	Cross-like Holey PnC Resonator	DIH-PnC Resonator
2 × 2	60,565	130,280	1,840,000
4 × 2	60,565	134,540	5,844,600
6 × 2	60,565	321,850	12,431,000
2 × 3	60,565	144,380	2,039,300
4 × 3	60,565	204,110	7,182,600
6 × 3	60,565	746,100	11,480,000
2 × 4	60,565	140,070	1,706,900
4 × 4	60,565	235,890	5,689,000
6 × 4	60,565	1,116,700	6,278,300

## Data Availability

All data needed to evaluate the conclusions in the paper are present in the paper.

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
