# Peer review of "Q-Factor Enhancement of Coupling Bragg and Local Resonance Band Gaps in Single-Phase Phononic Crystals for TPOS MEMS Resonator"

_micromachines, 2022, doi:10.3390/mi13081217_

Round 1
Reviewer 1 Report
In this manuscript, Li et al. designed a type of Double "I" Hole-phononic crystal with a single-phase coupling mechanism, which has ultra-wideband performance and strong attenuation. The position of the local resonance band gap can be adjusted by adjusting the width of the inner hole, which can achieve the coupling between the local resonance and Bragg mechanism. Applying this structure to the TPOS resonator, the anchor loss can be reduced effectively and the quality factor can be improved significantly with the smaller number of periodic arrays of phononic crystals. This manuscript is generally well-written and the results sound. Recommend publishing after addressing the following issues.
1. There is a general agreement in the field to use the term “phononic crystal” for band gaps based on Bragg scattering, and the term “acoustic metamaterial” or “elastic metamaterial” for local resonance band gaps. Therefore, I suggest that the authors use the term "acoustic metamaterial" in the introduction to describe the local resonance mechanism.
2. Add the keywords, such as anchor loss and MEMS resonator attenuation into the title of the manuscript to present the main contribution more specifically. “MEMS” is too broad.
3. At the end of the second paragraph of the introduction, when it is concluded that the phononic crystal occupies a large area in the TPOS resonator, a supplementary description of the reference is added.
4. Some minor issues: In Figure 8, the dimensioning of the support beam is too small, and a partially enlarged view can be added.
In the description of the x-axis and the y-axis in the legend of Fig. 11, the x-axis and the y-axis are not found in the figure. Therefore, I propose to supplement the corresponding coordinate system in Fig. 11
Author Response
请参阅附件。

Reviewer 2 Report
The authors propose a single-phase DIH-PnC structure, which provides the band gap based on the local resonance mechanism. The focus is on the transmission of the quality factor increase compared to other configurations due to the introduction of a single-phase DIH-PnC structure.
Which applications of the proposed PnC are expected? The latter should be described in detail. Many papers with numerical estimations of PnC have been presented recently, so the authors must describe the novelty of the configuration and specify possible applications.
Only 2-4 unit-cells in a row are used, so the boundary conditions used at the side boundaries of the specimen are important. Please describe which boundary conditions were employed for the sides of the specimen shown in Fig. 4.
The notations for wave vectors used in Eq. 1 and in the text are different. Besides, vectors are suggested to be different from scalar functions.
In the manuscript, sizes are given in um (at least in my PDF reader shows as um, e.g. a=24um, height ℎ=10um, hole m = 11um etc.) Should it be micrometres? If so, the latter must be corrected.
Author Response
请参阅附件。

Round 2
Reviewer 2 Report
The authors have improved the manuscript so that it is recommended for publication.